# The Evaluation of Healing Properties of *Galium verum*-Based Oral Gel in Aphthous Stomatitis in Rats

**DOI:** 10.3390/molecules27154680

**Published:** 2022-07-22

**Authors:** Miona Vuletic, Vladimir Jakovljevic, Suzana Zivanovic, Milos Papic, Mirjana Papic, Rasa Mladenovic, Vladimir Zivkovic, Ivan Srejovic, Jovana Jeremic, Marijana Andjic, Aleksandar Kocovic, Jasmina Sretenovic, Slobodanka Mitrovic, Biljana Božin, Nebojša Kladar, Sergey Bolevich, Jovana Bradic

**Affiliations:** 1Department of Dentistry, Faculty of Medical Sciences, University of Kragujevac, 34000 Kragujeva, Serbia; miona91kg@gmail.com (M.V.); suzanazivanovic91@yahoo.com (S.Z.); milos_papic@live.com (M.P.); mira.radovic@hotmail.com (M.P.); rasa.mladenovic@med.pr.ac.rs (R.M.); 2Department of Physiology, Faculty of Medical Sciences, University of Kragujevac, 34000 Kragujevac, Serbia; drvladakgbg@yahoo.com (V.J.); vladimirziv@gmail.com (V.Z.); ivan_srejovic@hotmail.com (I.S.); drj.sretenovic@gmail.com (J.S.); 3Department of Human Pathology, 1st Moscow State Medical University I.M. Sechenov, 119991 Moscow, Russia; bolevich2011@yandex.ru; 4Department of Pharmacy, Faculty of Medical Sciences, University of Kragujevac, 34000 Kragujevac, Serbia; jovana.jeremic@medf.kg.ac.rs (J.J.); andjicmarijana10@gmail.com (M.A.); salekkg91@gmail.com (A.K.); 5Department of Pathology, Faculty of Medical Sciences, University of Kragujevac, 34000 Kragujevac, Serbia; smitrovic@medf.kg.ac.rs; 6Department of Pharmacy, Faculty of Medicine, University of Novi Sad, 21000 Novi Sad, Serbia; biljana.bozin@mf.uns.ac.rs (B.B.); nebojsa.kladar@mf.uns.ac.rs (N.K.); 7Center for Medical and Pharmaceutical Investigations and Quality Control, University of Novi Sad, 21000 Novi Sad, Serbia

**Keywords:** recurrent aphthous stomatitis, *Galium verum* L., oral ulcer healing, phytotherapy, oxidative stress

## Abstract

Although oral ulcers represent one of the most frequent oral mucosal diseases, the available treatment is not sufficient to provide complete ulcer recovery without side-effects. Therefore, the aim of our study was to prepare a mucoadhesive oral gel based on *Galium verum* ethanol extract (GVL gel) and reveal its healing effects in the model of aphthous stomatitis in rats. Rats with oral ulcers were divided into the following groups: control (untreated), gel base (ulcer was treated with the gel base, three times per day for 10 days), and GVL gel group (the ulcer was treated with GVL gel in the same way as the gel base). Animals from each group were sacrificed on days 0, 3, 6, and 10 for collecting blood and ulcer tissue samples. Healing properties of oral gel were determined by clinical evaluation, as well as biochemical and histopathological examinations. Our findings suggest a significant decrease in the ulcer size in GVL gel group, with healing effects achieved through the alleviation of oxidative stress, reduction in COX-2 immunopositivity, and increase in collagen content in buccal tissue. Significant ulcer repairing potential of GVL gel highlights this oral mucoadhesive gel as a promising tool for prevention and treatment of RAS.

## 1. Introduction

Recurrent aphthous stomatitis (RAS) is the most common oral ulcer disease presented as round ulceration with an erythematous band and white-yellowish pseudomembranous center [1]. The ulcers are painful and typically localized on the tongue and buccal and labial mucosa, while their appearance on the keratinized palatal and gingival mucosa is less frequent [2]. Several factors have been identified as possible causes for RAS: genetic factors, local trauma, microbial factors, immunologic factors, allergy to some dietary constituents, nutritional factors, and psychosocial stress. However, the definitive etiology of RAS is still unknown, but immunologically mediated mechanisms that drive the pathogenesis of RAS are the most common [3]. Previous studies showed that pro-oxidant levels are significantly higher in patients with RAS, thus indicating a possible important role of oxidative stress in the development and progression of aphthous lesions [4]. In most cases, RAS is manifested as minor ulcers that are less than 10 mm in diameter and heal in 10–14 days without scars [5]. Management of RAS depends on the severity and frequency of the lesions with the main goal to promote healing and pain relief. Local therapy is usually sufficient; nevertheless, sometimes patients with major RAS or with many minor ulcers require systemic therapy. Currently available therapeutic strategies include antiseptics (e.g., chlorhexidine gluconate), antibiotics, local anesthetic gels, corticosteroids (e.g., prednisone), laser therapy, nonsteroidal anti-inflammatory drugs (e.g., pentoxifylline-PTX), and/or systemic immunomodulators (e.g., thalidomide) [6,7,8].

Despite the fact that corticosteroids and antibiotics are the most effective and recommended in the management of RAS, certain patients do not respond adequately to the treatment, and, more importantly, their long-term use is associated with serious side-effects. Great concern in RAS management is referring to the multiple resistance of microorganisms to available antimicrobials especially in diabetic and cancer patients at risk for poorly healing infected ulcers. Therefore, scientific attention in recent years has been oriented on exploring alternative methods such as phytotherapy. For years, phytotherapy has been used as an approach in RAS treatment since plant-based products contain bioactive compounds with pharmacological activity comparable to currently available synthetic drugs but with better tolerability [9]. Numerous data suggest that medical plants with anti-inflammatory, antibacterial, and antioxidant activities could be used for reducing the pain and healing time of RAS [10]. *Galium verum* L. (*G. verum*, lady’s bedstraw) belongs to the Rubiaceae family and, throughout history, has been known as an active herbaceous plant with multiple therapeutic properties. The effectiveness of this plant is based on several isolated bioactive complexes: iridoid glycosides, phenols, triterpenes, anthraquinones, etc. [11]. Previous phytochemical analysis suggested the presence of chlorogenic, caffeic, and 2-*p*-coumaric acids, as well as quercitrin and rutin, in *G. verum* aerial part extracts, and those compounds have been highlighted to mostly contribute to the overall therapeutic potential of this plant species. However, the chemical content of the extract is strongly dependent on the harvesting time, method of extraction, applied solvent, etc. [11,12]. To the best of our knowledge, there are no data referring to the role of this plant species in RAS management. Since previous studies confirmed that *G. verum* extracts possess strong antioxidant and antimicrobial activity, we hypothesized that topically applied extract would accelerate the healing of aphthous stomatitis.

Regarding the abovementioned data, the aim of this study was to formulate a mucoadhesive gel based on *G. verum* extract and explore its efficacy in healing of RAS in rats, with a special emphasis on the role of oxidative stress. The special novelty of our research is in providing an understanding of the influence of a *G. verum*-based topical product as an innovative therapeutic approach in RAS, which might be implemented alone or in combination with currently available pharmacological agents.

## 2. Materials and Methods

### 2.1. Plant Material

The plant *G. verum* was collected on 5 July 2021 in Dobroselica, on the southern precipice of Mt Zlatibor, Serbia. Identification of the plant material was performed at the Institute of Botany and Botanical Garden “Jevremovac”, University of Belgrade, Serbia. The collected plant was dried under dark conditions and powdered (sieve 0.75). Ethanol extract was prepared using the heat reflux extraction method with 100 g of the aerial part of the plant and 500 mL of 70% ethanol as the solvent, at a temperature of 60 °C for 2 h. Afterward, the extract was dried under reduced pressure by a rotary evaporator (RV05 basic; IKA, Bitterfeld-Wolfen, Germany) in order to obtain dry extract [13,14]. The choice of reflux extraction method in the current work was based on its well-known advantages compared to other conventional techniques such as maceration and percolation in terms of less time for obtaining extract and a higher concentration of isolated polyphenol. This method, especially in combination with a polar extraction solvent such as ethanol, has been reported as relevant and convenient for obtaining the extract with the highest yield and significant amount of polyphenols [14,15].

### 2.2. Phytochemical Analysis of G. verum Extract

A previously described analytical method based on high performance liquid chromatography (HPLC-DAD, HPLC 1100, Agilent Technologies, Waldbronn, Germany) was applied for quantitative and qualitative chemical profiling of the obtained extract. Specifically, the extracts were analyzed for the amounts of gallic, caffeic, *trans*-cinnamic, *p*-coumaric, chlorogenic, rosmarinic, and ferulic acid, as well as quercetin, rutin, and quercitrin. The compounds of interest were separated on a Nucleosil C18 column (250 mm × 4.6 mm, 5 μm particle size) held at 30 °C. The mobile phase consisting out of 1% (*v*/*v*) aqueous HCOOH (A) and methanol (B) was delivered in gradient mode according to the following program: 0 min 10% B, 10 min 25% B, 20 min 45% B, 35 min 70% B, 40 min 100% B, and 46 min 10% B, while the flow rate was variable (0–10 min, 1 mL/min; 10–20 min, 0.8 mL/min; 20–30 min, 0.7 mL/min; 30–46 min 1 mL/min). The elution of compounds from the column was monitored at 280 nm (gallic, caffeic and *trans*-cinnamic acid), 330 nm (*p*-coumaric, chlorogenic, rosmarinic, and ferulic acid, as well as quercetin), and 350 nm (rutin and quercitrin), while the quantification was based on calibration curves obtained for analyzed chemical standard substances under the same experimental conditions [16].

### 2.3. Preparation of G. verum Extract-Based Mucoadhesive Oral Gel

A simple gel base was prepared using Carbomer 934, propylene glycol, triethanolamine solution, and water as previously described by our research group [17]. Sodium benzoate as a preservative was dissolved in 40 °C purified water. Afterward, a defined amount of carbomer 934 was mixed with water solution until a homogeneous dispersion was formed using a magnetic stirrer (1200 rpm for 30 min). Triethanolamine solution (7 g of 10% solution) was added to obtain the gel base with a pH of ~6. The tested formulation, i.e., the gel based on *G. verum* extract, was prepared using 20% *w*/*w* of extract, which was incorporated into the carbomer gel to achieve a uniform gel [17,18].

### 2.4. In Vivo Oral Ulcer Healing Examinations

#### 2.4.1. Animals

Sixty male Wistar albino rats (210–270 g) obtained from the Military Medical Academy, Belgrade, Serbia, were included in the experimental study. The animals were housed at a temperature of 22 ± 2 °C, with 12 h of automatic illumination daily, and they consumed commercial rat food (20% protein rat food; Veterinary Institute Subotica, Serbia) ad libitum. The experimental study was carried out in the laboratory for cardiovascular physiology of the Faculty of Medical Sciences, University of Kragujevac, Serbia. The Ethics Committee of the Faculty of Medical Sciences (Number 01-205, January 2022) approved the experimental protocol. All procedures were performed according to the EU Directive for the welfare of laboratory animals (86/609/EEC) and the principles of Good Laboratory Practice (GLP).

#### 2.4.2. Induction of RAS Model

To create the oral buccal ulcer model, rats were anesthetized with a mixture of ketamine (5 mg/kg) and xylazine (10 mg/kg) intraperitoneally. Oral mucosal ulcer was performed by exposing the buccal surfaces to glacial acetic acid following a previously established acetic-acid-induced in vivo model in rats [19]. Two days later, chronic ulceration with well-defined borders developed, which was designated as day 0.

#### 2.4.3. Experimental Animals

Immediately after induction of RAS, rats were treated by applying the test formulation three times a day. Depending on the applied formulation, the rats were divided into three groups (20 per group): negative control (ulcer left without any treatment), control group—gel base (ulcer treated with gel base not containing any active agent), and GVL gel (ulcer treated with 20% *G. verum*-based gel).

The test formulation was topically applied using a cotton swab on a plastic stick for the gel (0.5 g) three times daily for 10 days. The animals were sacrificed at different time intervals for 10 days. Five animals from each group were sacrificed on days 0, 3, 6, and 10. After short-term ketamine/xylazine anesthesia, rats were sacrificed by decapitation. Blood was collected for determination of systemic redox status, while buccal tissues were taken for histopathological analysis and determination of tissue redox state.

#### 2.4.4. Estimation of Oral Buccal Ulcer Healing

The development and healing of the lesion were photographed and followed every second day by measuring the size of the ulceration using Image J software (v. 1.53; National Institutes of Health, Bethesda, MD, USA) starting from day 0 until complete healing. The degree of healing was represented as the percentage contraction of ulcer area, and it was calculated for each animal using the following formula: ulcer healing rate (%) = (A0 − At) × 100/A0, where A0 and At represent the initial ulcer area and the ulcer area at the time of observation, respectively [20].

### 2.5. Biochemical Analysis

#### 2.5.1. Evaluation of Systemic Redox State

In the moment of sacrificing animals, blood samples were collected from the jugular vein for biochemical analysis. The concentrations of pro-oxidants were determined in plasma: the levels of superoxide anion radical (O_2_^−^), nitrites (NO_2_^−^), and hydrogen peroxide (H_2_O_2_), as well as the index of lipid peroxidation (measured as thiobarbituric acid reactive substances (TBARS)). Parameters of the antioxidative defense were determined in the lysate of erythrocytes: activities of superoxide dismutase (SOD), as well as levels of reduced glutathione (GSH) and catalase (CAT).

##### Determination of the Values of Pro-Oxidants in Plasma Samples

The degree of lipid peroxidation in the sample was estimated by measuring TBARS using 1% TBA (thiobarbituric acid) in 0.05 NaOH, incubated in plasma at 100 °C for 15 min, and read on 530 nm spectrophotometrically. The level of NO_2_^−^ was measured, using Griess’s reagent, as an index of nitric oxide (NO_2_^−^) production, as previously described. Nitrites were measured at wavelength of 550 nm. The levels of superoxide anion radical (O_2_^−^) were measured using the nitro blue tetrazolium (NBT) reagent in Tris buffer (assay mixture) with a plasma sample. The measurements were performed at 530 nm. For measurement of hydrogen peroxide (H_2_O_2_), we used a protocol based on the oxidation of phenol red by hydrogen peroxide in the presence of horseradish peroxidase. The level of H_2_O_2_ was measured at 610 nm [17].

##### Determination of the Values of Antioxidants in Erythrocyte Samples

The level of reduced glutathione was determined on the basis of GSH oxidation with 5.5-dithio-bis-6.2-nitrobenzoic acid. The measurement was performed at 420 nm. CAT activity was determined according to previously described method, and detection was performed at 360 nm. For determination of SOD activity, the epinephrine method was used, and detection was performed at 470 nm [17].

#### 2.5.2. Evaluation of Redox State in Aphthous Lesions

After sacrificing animals, the aphthous lesions were isolated and immediately frozen at −80 °C. A 0.5 g section of each tissue was homogenized in 5 mL of ice-cold phosphate-buffered saline (pH 7.4). Afterward, the homogenate was centrifuged at 10,000× *g* at 4 °C for 15 min. The supernatants were collected and used for oxidative stress analysis. The following parameters were spectrophotometrically determined in supernatants: TBARS, GSH, CAT, and SOD, as previously described [19].

### 2.6. Histologic Analysis

The samples of isolated aphthous lesions were fixed in 4% formalin for 24 h. Afterward, tissue sections were dehydrated in increasing concentrations of alcohol (70–100%), cleared in xylene, and embedded in paraffin. Then, 5 μm thick tissue sections were stained with hematoxylin and eosin (H&E) in order to verify morphological changes, as well as with Picro-Sirius Red for detection of collagen. The specimens were analyzed using the following pre-established histological scoring protocol [21]:Presence of epithelial necrosis, no signs of inflammation;The inflammatory reaction has started, with no new capillary proliferation;The inflammatory reaction is prominent with few capillary proliferations on the basis of the ulcer, but no epithelization at the surface;The inflammatory reaction is decreased, new capillary proliferation has reached the surface, and epithelization has started at the surface;The epithelization is complete.

COX-2 in this study was verified by immunohistochemical staining. Tissue sections were deparaffinized and rehydrated through decreasing ethanol concentrations (100%–70%), and then sections were cooked in citrate buffer at a temperature of 95 °C, at pH 6.0, for 20 min. Tissue sections were cooled at room temperature for 15 min, followed by blocking endogenous peroxidase (into 3% hydrogen peroxide) for 10 min and washing in phosphate-buffered saline (PBS) for 5 min. Sections were incubated with protein block (Ultravision Protein Block, Thermo Scinetific, Waltham, MA, USA) for 5 min, followed by primary antibody COX-2 (1:100) (Novocastra, Leica Biosystems, Newcastle Upon Tyne, UK) for 1 h at room temperature. Sections were washed in PBS (3 × 5 min) followed by incubation with primary antibody amplifier Qvanto, Thermo Scientific, USA for 10 min, washed in PBS (5 min) followed by addition of HRP (HRP Polymer Quanto) for 10 min, and then washed in PBS for 5 min. Binding sites were visualized with 0.05% diaminobenzidine (DAB; Serva, Heidelberg, Germany). The sections were contrasted with hematoxylin, rehydrated in increasing alcohol concentrations (70–100%), cleared in xylene, and mounted with DPX (Sigma-Aldrich, St. Louis, MO, USA) [22]. Images of the tissue sections were captured using a digital camera attached to a light microscope Olympus BX51 (Olympus Life and Material Science Europa GmbH, Hamburg, Germany). Morphometric analysis of collagen content and immunopositivity evaluation of the COX-2 in the aphthae were performed using the software program Image Pro-Plus (Media Cibernetics, Rockville, MD, USA) according to previously described methodology [22,23]. Results are presented as percentages [22,23].

### 2.7. Statistical Analysis

Statistical analysis was performed using the Statistical Package for Social Sciences v23.0 (SPSS; IBM Corp., Armonk, NY, USA). The Shapiro–Wilk test was used to evaluate the distribution of data. Parametric tests (one-way ANOVA and independent-samples *t*-test) and nonparametric tests (Kruskal–Wallis H and Mann–Whitney U) were performed to determine the differences between groups depending on the normal distribution of the data. All results are expressed as the mean value ± standard deviation (SD). A *p*-value < 0.05 was considered statistically significant.

## 3. Results

### 3.1. HPLC-DAD Analysis of G. verum Ethanol Extract

The chemical composition of the extract analyzed by HPLC-DAD is shown in Table 1 and Figure 1. Several bioactive compounds were detected; however, the most abundant compounds were rutin, *p*-coumaric acid, quercetin, quercitrin, and gallic, caffeic, and chlorogenic acid, while ferulic, *trans*-cinnamic, and rosmarinic acid were below the limit of detection.

### 3.2. Ulcer Healing Rate

The role of *G. verum* extract-based gel in oral buccal ulcer healing was investigated during the 10 days of application. Therefore, the ulcer area of the rats in each group was measured every second day starting from day 0. As illustrated in Figure 2 and Figure 3, application of GVL gel resulted in a significant increase in the percentage ulcer contraction compared to the gel base and negative control group, on all key days throughout the duration of the experiment. The beneficial effect of GVL gel was obvious from the second day of treatment, whereby the ulcer contraction of 61% in this group was significantly higher in comparison to other examined groups. This trend continued throughout the duration of the treatment; accordingly, therefore on the sixth day, we noticed ulcer contraction in this group of 100%, representing a twofold higher percentage contraction in comparison to the control groups (Figure 2 and Figure 3).

### 3.3. Redox Status

#### 3.3.1. Systemic Redox Status

There were no significant differences in the levels of pro-oxidants and antioxidants in systemic circulation between groups (*p* > 0.05; Figure 4 and Figure 5).

#### 3.3.2. Redox Status in Oral Buccal Ulcer Lesion

The level of TBARS, on days 3 and 6, was decreased in GVL group compared to the negative control and gel base groups (*p* < 0.05; Figure 6). There were no statistically significant differences between the values of TBARS in the gel base and negative control groups on any day of observation (*p* > 0.05; Figure 6).

A significantly higher level of GSH and CAT activity in buccal tissue was found in the GVL gel group on days 3 and 6, compared to the gel base and negative control groups (*p* < 0.05; Figure 6). Moreover, SOD activity was increased in the GVL gel group on day 6 compared to gel base and negative control groups (*p* < 0.05; Figure 6). There were no significant differences between the gel base and negative control groups in the values of all observed markers on days 3, 6, and 10 (*p* > 0.05; Figure 6).

### 3.4. Histological Analysis of Oral Buccal Ulcer Lesion

Histological evaluations of oral buccal ulcer samples on days 0, 3, 6, and 10 are presented in Table 2. Microscopic analysis of the samples in the buccal mucosa of rats revealed epithelial necrosis, subendothelial acute inflammatory infiltrate with predominance of neutrophils, and part ulceration with dilated vessels in all groups on day 3. A moderate chronic inflammatory process with fibroblastic proliferation and organized collagen fibers was observed in the gel base group and negative control on day 6, while remodeled connective tissue and the absence of ulcer were noted in the group treated with GVL gel. Additionally, the healing score for the GVL gel was statistically significantly higher than that for the other groups on day 6 (Table 2, Figure 7). Healing, characterized by the presence of connective tissue, was observed on day 10 in the control and gel base groups, with healing score reaching similar values to the GVL group (Table 2, Figure 7).

COX-2 immunopositivity increases during inflammation, while the healing process is characterized by a reduction in COX-2 immunopositivity. In our study, there was no difference in expression of COX-2 on day 0; however, on days 3 and 6, a significant difference between the GVL group and untreated ulcerated rats was observed (*p* < 0.05, Figure 8 and Figure 9). The COX-2 immunopositivity on day 10 was similar in groups of animals treated with the gel base and control groups. On the other hand, treatment with GVL gel significantly decreased the immunopositivity of COX-2 compared to other groups (*p* < 0.05, Figure 8 and Figure 9).

There was no difference between groups in terms of the collagen content on day 0. Nevertheless, treatment with GVL-based gel significantly increased the production of collagen, on days 3 and 6, in relation to the gel base and control group (*p* < 0.05; Figure 10).

## 4. Discussion

Currently available therapy of RAS with antibiotics and corticosteroids does not provide complete ulcer cure, leads to serious side-effects, and increases the risk for microbial resistance and, consequently, unsuccessful buccal mucosa regeneration. Therefore, there is an urgent need for the identification of safe, potent, and cost-efficient novel oral ulcer healing agents [24,25,26,27]. Current trends for managing oral wounds and related complications highlight phytotherapy as a well-known alternative to synthetic drugs. Medicinal plants contain various compounds with proven antioxidant, antimicrobial, and anti-inflammatory properties that would accelerate oral wound healing. Polyphenols, as one of the most valuable molecules in plant extracts, have been widely screened for their oral wound-healing potential [28,29]; however, there are still numerous plants that need to be investigated in order to find a novel efficacious therapeutic approach that would increase patients’ life quality. The choice of *G. verum* as the active component in our study was based on previous investigations by our laboratory team, as well as other studies that confirmed the presence of strong antioxidant, anti-inflammatory, and antimicrobial constituents in this plant species [13,30]. The special novelty of our study is in examining a completely uninvestigated plant species in oral ulcer management. Moreover, we used a biocompatible gel based on *G. verum* ethanol extract to provide a longer retention time on buccal mucosa, consequently achieving greater efficacy in comparison to liquid drug delivery forms.

In the first part of our research, chemical analysis of the ethanol extract of *G. verum* confirmed the presence of compounds such as rutin and *p*-coumaric acid in the largest amount, while significant amounts of quercetin, quercitrin, and gallic, caffeic, and chlorogenic acid were also detected. In accordance with our research, the main ingredient in the 70% ethanolic extract of *G. verum* in the investigation conducted by Vlase was rutin [12,31]. Moreover, *p*-coumaric acid, quercetin, and quercitrin were also detected but in lower amounts. Our results are partially in agreement with the literature data suggesting these phenolic compounds as the major constituents of *G. verum.* However, qualitative and quantitative differences between the composition of our extract and others can be attributed to the different geographical origins of the plant material. Moreover, chemical content is strongly affected by the applied method of extraction and extraction solvents [31,32]. Gel with carbomer was chosen as the drug delivery system in our study since the mucoadhesive properties of this polymer enable prolonged contact at the site of administration and present a comfortable preparation for application to oral ulcers [33].

After preparation of the topical formulation, we aimed to assess its efficacy on an in vivo rat model of chemically induced aphthous stomatitis through clinical, biochemical, and histological examinations. We focused on a clinical evaluation of the healing effects of GVL gel by estimating the duration for healing, as well as performing ulcer size measurements and calculating ulcer contraction. Our findings revealed that topical therapy with GVL gel significantly contributed to oral buccal healing, which was verified by the lowest average ulcer healing time, as well as the most prominent reduction in ulcer size in comparison to other groups. In the present study, a decrease in ulcer area was observed day by day in all groups; however, the dynamics of the recovery was different among rats treated with GVL gel and untreated rats. The ulcers were almost closed after 4 days of GVL gel administration, while, in untreated rats and rats treated with gel base, the ulcers were still notable in all samples. The greatest percentage ulcer contraction in rats with topical GVL gel administration indicates the potential of this formulation to significantly accelerate the recovery of the buccal mucosa. Our results are in accordance with previous studies suggesting that the healing time of chemically induced oral ulcer is around 10 days without treatment and 4–6 days after treatment with plant extracts containing polyphenols [7,34,35]. Literature data report the great capability of caffeic acid, *p*-coumaric acid, quercetin, luteolin, etc. in healing of oral ulcers [36]. Therefore, the shorter time for complete ulcer healing in rats treated with GVL gel in our research might be attributed to the action of bioactive constituents found in the high quantities in our extract such as rutin, *p*-coumaric acid, quercetin, quercitrin, and gallic, caffeic, and chlorogenic acid.

Numerous data support the fact that oxidative stress is involved in the progression of RAS, thus suggesting that future drugs should act on a reduction in the harmful effects of reactive oxygen species (ROS). Elevated levels of pro-oxidant markers in patients with active oral lesions in relation to patients in the remission stage of RAS and healthy controls have been well documented [37,38]. Therefore, we hypothesized that *G. verum* extract, as a strong antioxidant agent formulated in gel, may accelerate the healing process in oral mucosa via attenuating oxidative stress. Since a proper understanding of redox homeostasis is essential for finding an optimal strategy, we aimed to assess the impact of GVL gel on both systemic and local redox status. Application of GVL did not change pro-oxidant or antioxidant markers in blood samples, thus indicating no systemic antioxidant effects of this formulation. The absence of alterations of systemic redox balance is expected since carbomer, as a mucoadhesive agent, possesses the ability to localize drug absorption, which leads to an elevated concentration of active compounds at the site of the ulcer.

In addition to biochemical analysis in blood samples, we monitored oxidative stress in the ulcer tissue in order to provide a better understanding of the effect of GVL gel. Our results revealed that application of GVL gel was associated with a significant reduction in TBARS levels in ulcer tissue, which was prominent on days 3 and 6 following ulcer appearance. Moreover, there were significant differences in all measured antioxidant parameters, which were increased in the GVL group compared to the gel base and control group, on days 3, 6, and 9. Such results indicate an important role of GVL gel on oxidative stress in terms of amplification of the antioxidative defense system and a reduction in the concentration of pro-oxidative markers. Several studies support our results showing increased activity of SOD and GSH in mucosal ulcer, by 40% and 54%, respectively, after treatment with plants with a similar phytochemical profile [39,40].

In order to highlight the mechanisms responsible for healing effects of GVL gel, histological analysis of the buccal mucosa was performed. H&E staining confirmed that GVL gel reduced the duration of the inflammatory process and accelerated healing time. The most prominent beneficial effects were achieved on day 6, which was confirmed by the completely remodeled epithelium and absence of ulcer in rats exposed to GVL treatment. On the contrary, the inflammatory response was still present in other groups at that timepoint. These results are in accordance with macroscopic evaluations, thus confirming the significant role of GVL in promoting ulcer healing.

It is well known that cyclooxygenase enzymes are essential in the process of inflammation and participate in the process of prostaglandin generation. Since COX-2 is an inducible enzyme that promotes inflammatory prostaglandins at sites of healing and inflammation, we aimed to assess the influence of GVL gel on COX-2 protein expression in buccal ulcer tissue [41]. Our findings suggested a significant decrease in COX-2 protein expression on days 3 and 6 in rats treated with GVL gel compared to other experimental groups. The reduction in COX-2 immunopositivity after GVL application indicates an alleviated inflammatory process, which is a crucial step in healing promotion. Previous research also showed decreased expression of COX-2 protein in mucosal ulcer after treatment with plants rich in polyphenols [42]. The anti-inflammatory potential of *G. verum*, as the active ingredient in our mucoadhesive gel, could be explained by the presence of phenolic acids such as *p*-coumaric, caffeic, and chlorogenic acid, which are known as selective COX-2 inhibitors [43]. Our results indicate that the healing capacity of GVL gel is partially mediated via the reduction in COX-2.

Taking into consideration that collagen contraction is the final stage in wound healing, after inflammation and cell proliferation [44], we aimed to reveal if GVL gel application may alter collagen content in buccal ulcer samples. Our results indicate the potential of GVL gel to elevate collagen content in comparison to other groups on days 3 and 6, thus contributing to buccal mucosa regeneration. Previous studies also showed that flavonoids and other polyphenols contribute to preservation of structural integrity of ulcer tissue [45,46]. We may assume that active compounds of the plant extracts in our formulation may increase the strength of collagen fiber, which leads to higher DNA synthesis, better circulation, and a reduction in cell damage [47].

## 5. Conclusions

Our results for the first time suggest the potential of a mucoadhesive gel based on *G. verum* extract to accelerate oral ulcer healing. The positive effects of this gel are reflected by a significant increase in percentage ulcer contraction throughout the duration of the treatment. Additionally, 10 days of treatment with *G. verum* gel reduced the generation of pro-oxidants in buccal tissue, thus suggesting a reduction in oxidative stress as one of the mechanisms underlying the healing effects of this formulation. The potential of the gel to decrease COX-2 expression and elevate collagen content further contributes to better buccal mucosa regeneration. This significant influence of *G. verum* extract-based gel on oral wound healing may be attributed to the additive and/or synergistic activity of the most abundant compounds in our extract such as rutin, *p*-coumaric acid, quercetin, quercitrin, and gallic, caffeic, and chlorogenic acid. The promising findings of our study highlight GVL as a promising tool for the management of oral mucosal ulceration.

## Figures and Tables

**Figure 1 molecules-27-04680-f001:**
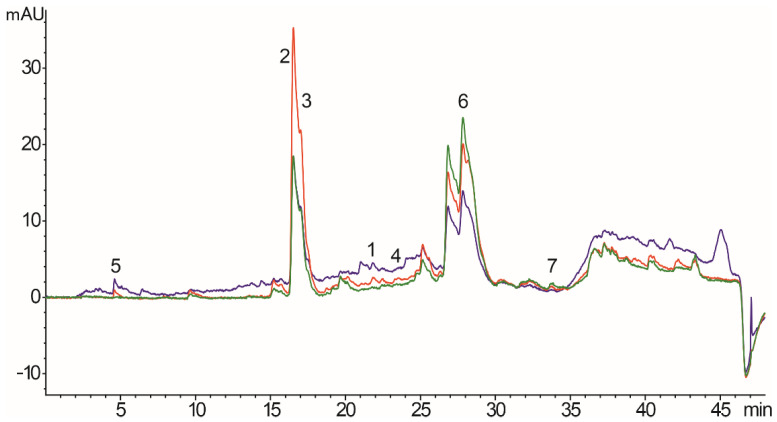
Chromatograms of *G. verum* ethanol extract with detection at 280 nm (blue line), 330 nm (red line), and 350 nm (green line). Identified compounds: 1—caffeic acid, 2—*p*-coumaric acid, 3—quercetin, 4—chlorogenic acid, 5—gallic acid, 6—rutin, and 7—quercitrin.

**Figure 2 molecules-27-04680-f002:**
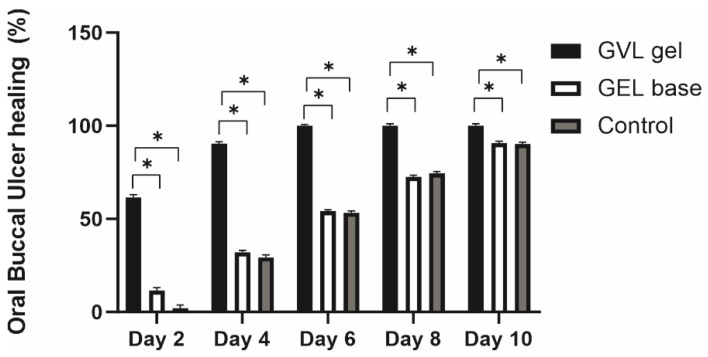
Effect of treatment with *G. verum*-based gel on rate of buccal ulcer healing. Values represent the means obtained with five animals in each group (*n* = 5 animals per group). * *p* ≤ 0.05 statistically significant difference between compared groups.

**Figure 3 molecules-27-04680-f003:**
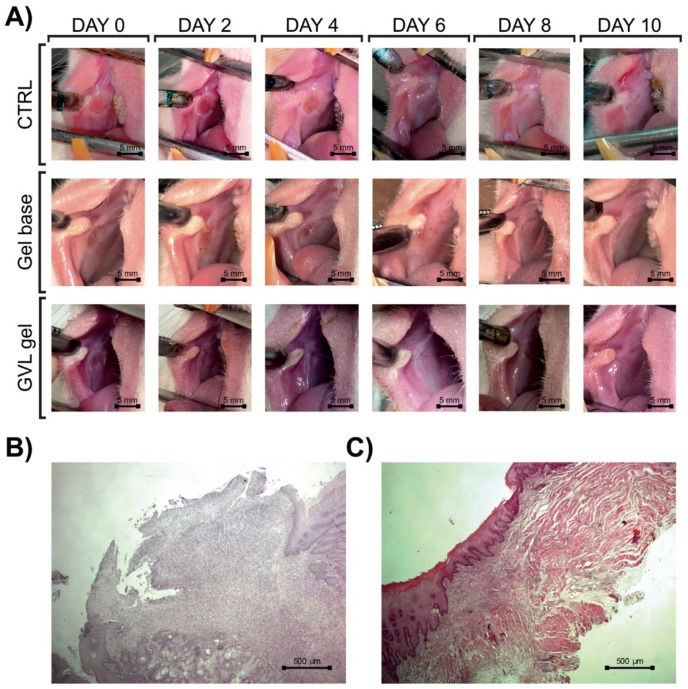
Impact of *G. verum*-based gel on (**A**) macroscopic characteristics of ulcer on different days (0, 2, 4, 6, 8, and 10), (**B**) microscopic characteristics of developed oral ulcer, and (**C**) microscopic characteristics of healed ulcer (*n* = 5 animals per group).

**Figure 4 molecules-27-04680-f004:**
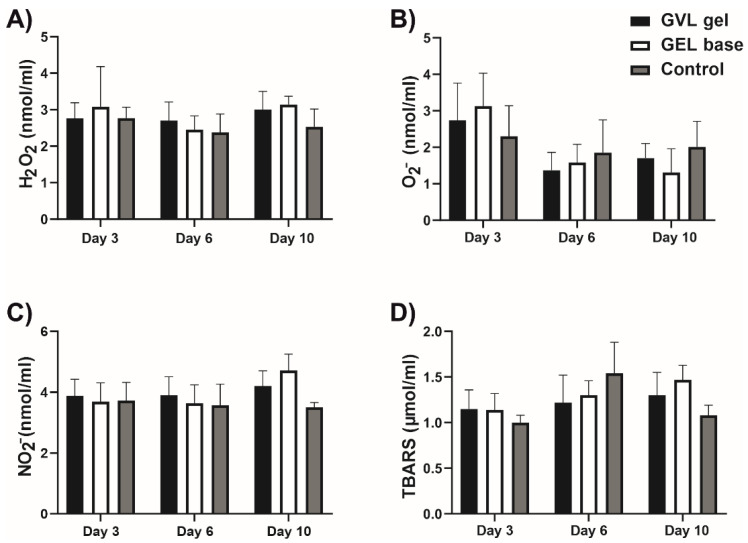
Effects of *G. verum*-based gel on pro-oxidative markers in systemic circulation on days 3, 6, and 10: (**A**) H_2_O_2_—hydrogen peroxide; (**B**) O_2_^−^—superoxide anion; (**C**) NO_2_^−^—nitric oxide; (**D**) TBARS—thiobarbituric acid-reactive substances. Values are expressed as the mean ± (SD); *n* = 5 animals per group.

**Figure 5 molecules-27-04680-f005:**
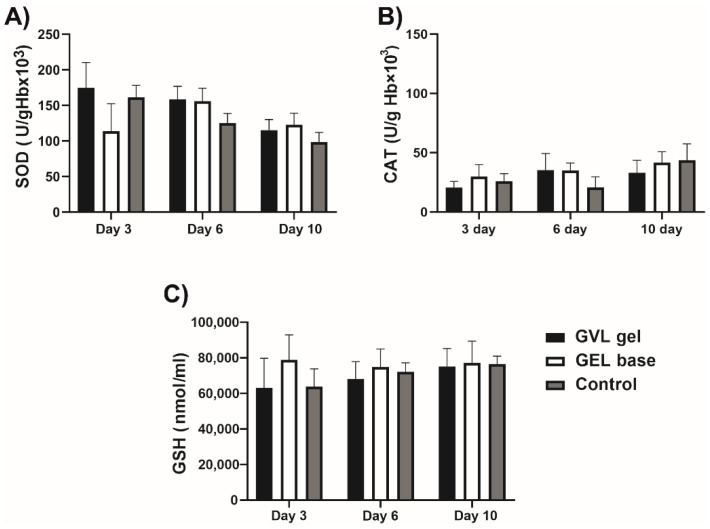
Effects of *G. verum*-based gel on antioxidative markers in systemic circulation on days 3, 6, and 10: (**A**) SOD—superoxide dismutase; (**B**) CAT—catalase; (**C**) GSH—reduced glutathione. Values are expressed as the mean ± (SD); *n* = 5 animals per group.

**Figure 6 molecules-27-04680-f006:**
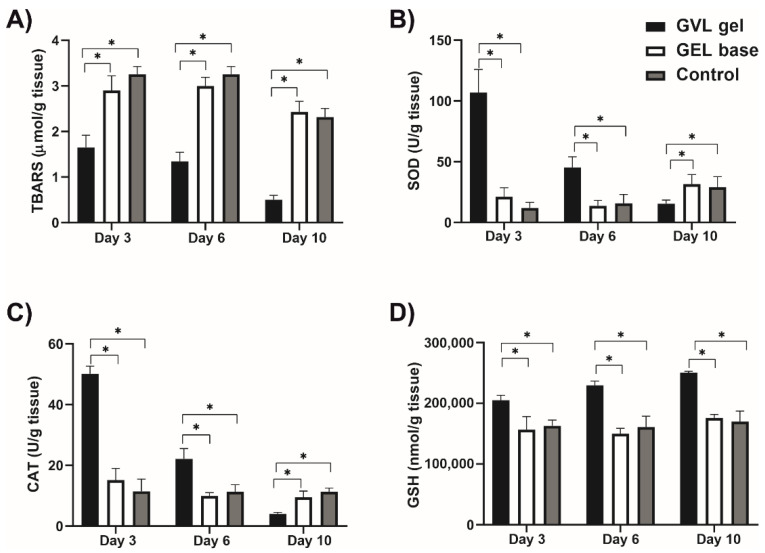
Effects of *G. verum*-based gel on pro-oxidative and antioxidative markers in oral buccal ulcer lesion on days 3, 6, and 10: (**A**) TBARS—thiobarbituric acid-reactive substances; (**B**) SOD—superoxide dismutase; (**C**) CAT—catalase; (**D**) GSH—reduced glutathione. Values are expressed as the mean ± (SD); *n* = 5 animals per group. * *p* ≤ 0.05 statistically significant difference between compared groups.

**Figure 7 molecules-27-04680-f007:**
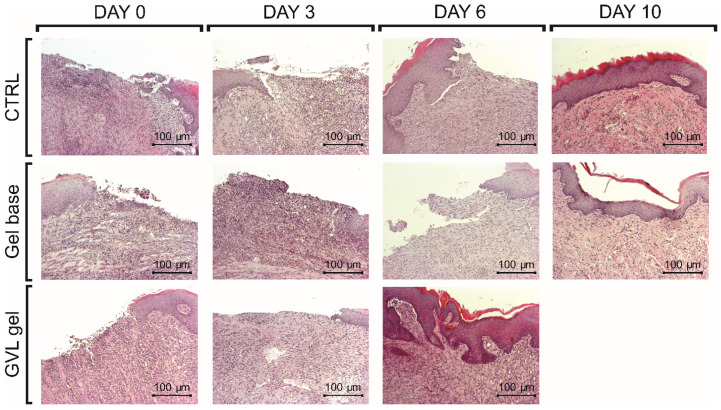
Representative photographs of H&E staining of the oral ulcer on days 0, 3, 6, and 10 (magnification 10×; scale bar = 100 µm).

**Figure 8 molecules-27-04680-f008:**
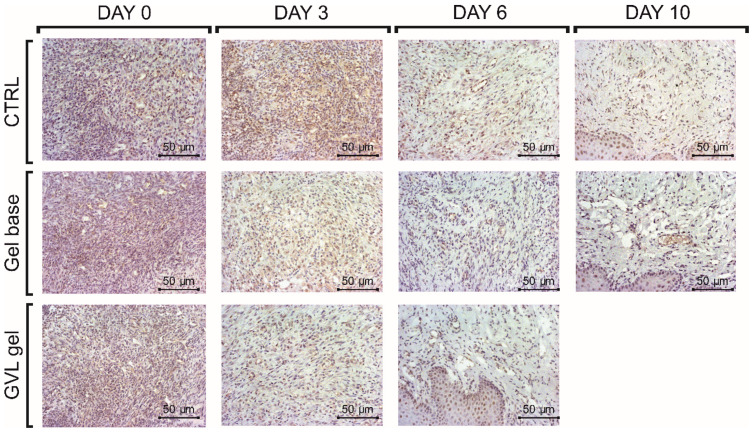
Representative photographs of COX-2 staining of the oral ulcer on days 0, 3, 6, and 10 (magnification 20×; scale bar = 50 µm); *n* = 5 animals per group.

**Figure 9 molecules-27-04680-f009:**
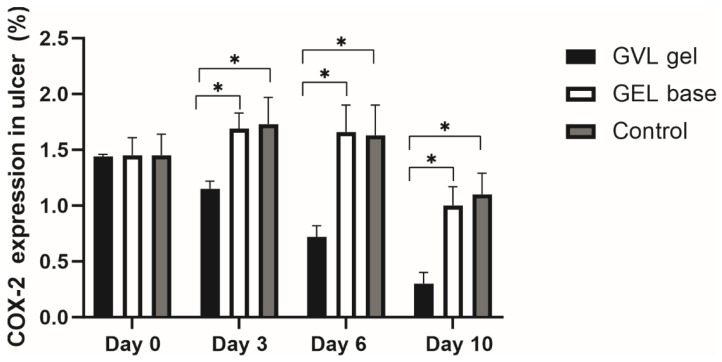
Effects of applied formulations on the COX-2 immunopositivity in oral ulcer. Values are expressed as the mean ± standard deviation; *n* = 5 animals per group. * *p* ≤ 0.05 statistically significant difference between compared groups.

**Figure 10 molecules-27-04680-f010:**
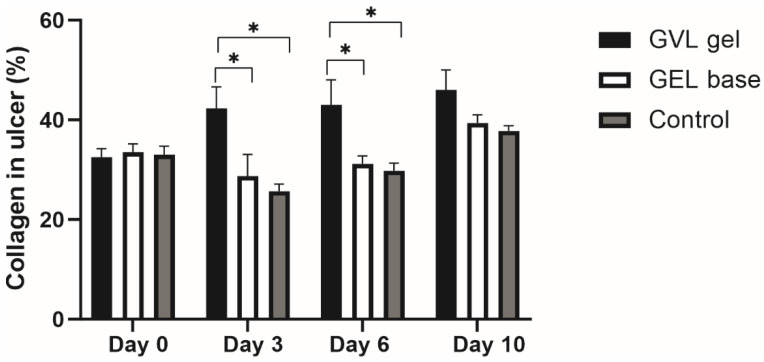
Effects of applied formulations on the collagen content in oral ulcer. Values are expressed as the mean ± standard deviation; *n* = 5 animals per group. * *p* ≤ 0.05 statistically significant difference between compared groups.

**Table 1 molecules-27-04680-t001:** Quantitative and qualitative analysis of individual compounds found in *G. verum* extract expressed as mg/g of dry extract.

Name of Compound	*G. verum* Ethanol Extract
Rutin	23.81 ± 1.90
*p*-Coumaric acid	9.35 ± 0.93
Quercetin	0.76 ± 0.05
Quercitrin	0.72 ± 0.04
Gallic acid	0.53 ± 0.08
Caffeic acid	0.12 ± 0.01
Chlorogenic acid	0.16 ± 0.01
Ferulic acid	<0.1
*trans*-Cinnamic acid	<0.03
Rosmarinic acid	<0.02

**Table 2 molecules-27-04680-t002:** Histological healing scores for each tested group on days 0, 3, 6, and 10.

	Day 0	Day 3	Day 6	Day 10
Control	2 ± 0.00	3 ± 0.00	4.16 ± 0.41	4.83 ± 0.41
GEL base	2.33 ± 0.82	3.16 ± 0.41	4.33 ± 0.52	5 ± 0.00
GVL gel	2.67 ± 0.51	4 ± 0.00	5 ± 0.00 *	5 ± 0.00

* Statistically significant difference at the level of *p* ≤ 0.05 compared to the control group.

## Data Availability

Data is contained within the article.

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
