# Peer review of "The Evaluation of Healing Properties of Galium verum-Based Oral Gel in Aphthous Stomatitis in Rats"

_molecules, 2022, doi:10.3390/molecules27154680_

Round 1
Reviewer 1 Report
This study by Vuletic et al discussed the healing properties of Galium verum, formulated in a buccal-adhesive gel on aphthous stomatitis ulcers induced in rats. Although the pharmacological research is interesting, the author claimed to study the pharmaceutical formulation and phyto-composition of the ethanolic extract of the plant, which I was not able to spot inside this manuscript. I also have some concerns about the experimental design. The details are as follows:
The Title
The title is confusing when it is read. I recommend the title be as follows:
“The healing properties of Galium verum-based oral gel in aphthous stomatitis in rats: formulation and evaluation”
The Abstract
The abstract should have a background statement
The Introduction
I recommend adding a paragraph about the major phytoconstituent of the plant to support the part of phyto-components identification and quantification in the manuscript.
Materials and methods:
Line 91: “….dehydrated…” I think dried will be better
Lines 92-93: why use heat reflux with ethanol extraction? And for how long?
Line 93: “….. condensed….” I prefer “dried under reduced pressure” because rotary evaporators do not condense the extract but condense the pure solvent.
Line 98: What are the 10 major compounds? and when checking reference [13], it did not show any relevance to the manuscript, revise
Section 2.2: What are the conditions of the LCMS-MS and the HPLC used?
Section 2.3: the section needs a reference. Furthermore, this section shows that the gel formulation of Galium verum was not studied. Thus, I recommend deleting the word “formulation” from the title.
Line 103: I have a concern about using triethanolamine in oral preparation. How much of it was used to adjust the pH?
Section 2.4.3: Why use 20 animals per group? And I wonder how the ethical committee approved this experimental design? Why sacrifice 5 animals to confirm the establishment of the ulcer, although, in the next section, you were able to photograph the ulcers daily? I have a major concern about the animal experimental design; the authors have to justify the design and procedures.
Line 136: “Five animals from each group will be ……” it should be in the past tense.
Lines 205 to 209: The methodology of Cox-2 immunohistochemistry should be more detailed and referenced
Results
Section 3.1: How were these compounds identified (in Figure 1) and quantified (Table 1)? Figure 1 is a DAD trace which is not enough to identify the compounds? Where is the LC-MS-MS trace mentioned in the methodology section? And why did you mention that there are the major 10 compounds, although the last 3 (according to table 1) are less than 0.1 mg/ gram extract?
Section 3.2: In section 2.4.4, you mentioned, “The development and healing of the lesion were photographed and followed daily by measuring the size of the ulceration using Image J software”; however, in this section (3.2), you mentioned:” Therefore, the ulcer area of the rats in each group was measured every second day starting from day 0” So which is right? Every day or every other day?
Discussion
Lines 369 to 371: “….Moreover, we developed a biocompatible gel based on G. verum ethanol extract….” You did not develop, because you did not make any pharmaceutical study on the gel formulation. The authors confirmed this in lines 377-380, where they confirm the reason for carbomer use in their gel. You just used a predetermined and predefined Gel formula. So the wording here needs to be changed to address this fact.
Lines 372-374: I did not understand until now how the phyto-compound separation, identification, and quantification study was performed. This part needs more clarification in the methodology and results and needs more discussion here.
Conclusion:
This part needs more attention to summarize all the study findings and define its significant application.
Other general notes:
The English language needs attention very much, especially the expression and grammar.
Author Response
Response to reviewers comments
This study by Vuletic et al discussed the healing properties of Galium verum, formulated in a buccal-adhesive gel on aphthous stomatitis ulcers induced in rats. Although the pharmacological research is interesting, the author claimed to study the pharmaceutical formulation and phyto-composition of the ethanolic extract of the plant, which I was not able to spot inside this manuscript. I also have some concerns about the experimental design. The details are as follows:
The Title
The title is confusing when it is read. I recommend the title be as follows:
“The healing properties of Galium verum-based oral gel in aphthous stomatitis in rats: formulation and evaluation”
Response: Thank you for this very useful comment. Taking into consideration this suggestion as well as suggestion of other reviewer mentioned below to remove the word ‘’formulation’’ from the title, we suggest this tittle: The evaluation of healing properties of Galium verum-based oral gel in aphthous stomatitis in rats. Please see Tittle page.
The Abstract
The abstract should have a background statement
Response: Thank you for this comment. We added a background statement in Abstract, please see abstract section.
The Introduction
I recommend adding a paragraph about the major phytoconstituent of the plant to support the part of phyto-components identification and quantification in the manuscript.
Response: Thank you for this comment. We expanded the introduction by adding a paragraph about the dominant bioactive compounds with their pharmacological effects, as you suggested. Please, see the Introduction section and Reference list (Reference 12).
Materials and methods:
Line 91: “….dehydrated…” I think dried will be better
Response: Thank you for this comment. The suggested word has been added in this section, please see Material and methods section, line 101.
Lines 92-93: Why use heat reflux with ethanol extraction? And for how long?
Response: The choice of reflux extraction method in the current work is based on its well known advantages compared to other conventional techniques such as maceration and perco-lation in terms of less time for obtaining extract and higher concentration of isolated polyphenols. This method especially in combination with ethanol as a sol-vent has been reported as suitable and convenient for reaching the extract with the highest yield and significant amount of polyphenols in extracts. The precise extraction conditions were added in the text, please see Lines 102-111.
Line 93: “….. condensed….” I prefer “dried under reduced pressure” because rotary evaporators do not condense the extract but condense the pure solvent.
Response: Thank you for this useful comment. We corrected suggested expression, please see Material and methods section, line 99.
Line 98: What are the 10 major compounds? and when checking reference [13], it did not show any relevance to the manuscript, revise
Response: Compounds identified in extract involved gallic, caffeic, trans-cinnamic acid, p-coumaric, chlorogenic, rosmarinic and ferulic acid, as well as quercetin, rutin and quercitrin. We apologize for the type error for reference 13, we inserted relevant reference for this part, please see Material and methods section, line 122 and reference number 16 in Reference list.
Section 2.2: What are the conditions of the LCMS-MS and the HPLC used?
Response: The chemical composition of the extract was determined by HPLC-DAD, NOT by LC-MS-MS, which was stated by mistake in the first version of manuscript. The compounds were identified by comparing the retention times and UV spectra of chemical standard substances analyzed, with the compounds eluting from the column – which is described in detail in one of our previous papers.
Section 2.3: The section needs a reference. Furthermore, this section shows that the gel formulation of Galium verum was not studied. Thus, I recommend deleting the word “formulation” from the title.
Response: The reference was provided in the text, please see Section 2.3. Additionally, the word ‘’formulation’’ was removed from the title.
Line 103: I have a concern about using triethanolamine in oral preparation. How much of it was used to adjust the pH?
Response: Triethanolamine was added according to previously published research where the authors also formulated oral gel for stomatitis in rats (please see reference 18). Another study also used this component for obtaining oral mucoadhesive gel as periodontal drug delivery. In our research, we added 7 g of 10% triethanolamine solution.
Aslani A, Ghannadi A, Najafi H. Design, formulation and evaluation of a mucoadhesive gel from Quercus brantii L. and coriandrum sativum L. as periodontal drug delivery. Adv Biomed Res. 2013 Mar 6; 2:21.
Aslani A, Zolfaghari B, Davoodvandi F. Design, Formulation and Evaluation of an Oral Gel from Punica Granatum Flower Extract for the Treatment of Recurrent Aphthous Stomatitis. Adv Pharm Bull. 2016 Sep; 6(3):391-398.
Section 2.4.3: Why use 20 animals per group? And I wonder how the ethical committee approved this experimental design? Why sacrifice 5 animals to confirm the establishment of the ulcer, although, in the next section, you were able to photograph the ulcers daily? I have a major concern about the animal experimental design; the authors have to justify the design and procedures.
Response: Thank you for this comment. The applied experimental model in this research is widely used in literature for assessing the impact of different topical formulations on oral ulcer healing. The number of animals per group (20) was calculated by using statistical power analysis program G*Power 3. The program calculates the sample size for animal studies based on pre-designed effect size at small, medium, and large difference between the groups based on Cohen's principles.The data obtained from previous researches with the similar experimental design and measured variables were used for G*Power 3 calculations (1, 2). Five animals from each group were sacrificed on day 0, 3, 6 and 10 day to precisely monitor and assess the impact of gel formulation on morphological changes in buccal tissue over time. Please see references below:
- Slomiany BL, Slomiany A. Biphasic role of platelet-activating factor in oral mucosal ulcer healing. IUBMB Life. 2003;55(8):483-90.
- Ayoub N, Badr N, Al-Ghamdi SS, Alsanosi S, Alzahrani AR, Abdel-Naim AB, Nematallah KA, Swilam N. HPLC/MSn Profiling and Healing Activity of a Muco-Adhesive Formula of Salvadora persica against Acetic Acid-Induced Oral Ulcer in Rats. Nutrients. 2021 Dec 22;14(1):28. doi: 10.3390/nu14010028. PMID: 35010903; PMCID: PMC8746813.
- Cavalcante GM, Sousa de Paula RJ, Souza LP, Sousa FB, Mota MR, Alves AP. Experimental model of traumatic ulcer in the cheek mucosa of rats. Acta Cir Bras. 2011;26(3):227-34.
- Brizeno LA, Assreuy AM, Alves AP, Sousa FB, de B Silva PG, de Sousa SC, Lascane NA, Evangelista JS, Mota MR. Delayed healing of oral mucosa in a diabetic rat model: Implication of TNF-α, IL-1β and FGF-2. Life Sci. 2016 Jun 15;155:36-47. doi: 10.1016/j.lfs.2016.04.033. Epub 2016 May 14. PMID: 27188585.
- Silva PGB, de Codes ÉBB, Freitas MO, Martins JOL, Alves APNN, Sousa FB. Experimental model of oral ulcer in mice: Comparing wound healing in three immunologically distinct animal lines. J Oral Maxillofac Pathol. 2018 Sep-Dec;22(3):444.
Line 136: “Five animals from each group will be ……” it should be in the past tense.
Response: Thank you for this comment. We corrected the sentence to past tense.
Lines 205 to 209: The methodology of Cox-2 immunohistochemistry should be more detailed and referenced
Response: Thank you for this comment. We described detailed methodology for Cox2 immunohistochemical staining and inserted reference for our methodology. Please see section methodology, part histologic analysis.
Results
Section 3.1: How were these compounds identified (in Figure 1) and quantified (Table 1)? Figure 1 is a DAD trace which is not enough to identify the compounds? Where is the LC-MS-MS trace mentioned in the methodology section? And why did you mention that there are the major 10 compounds, although the last 3 (according to table 1) are less than 0.1 mg/ gram extract?
Response: The chemical composition of the extract was determined by HPLC-DAD, NOT by LC-MS-MS, which was stated by mistake in the first version of manuscript. The compounds were identified by comparing the retention times and UV spectra of chemical standard substances analyzed, with the compounds eluting from the column – which is described in detail in one of our previous papers (reference 17).
Section 3.2: In section 2.4.4, you mentioned, “The development and healing of the lesion were photographed and followed daily by measuring the size of the ulceration using Image J software”; however, in this section (3.2), you mentioned:” Therefore, the ulcer area of the rats in each group was measured every second day starting from day 0” So which is right? Every day or every other day?
Response: Thank you to noticing that. We apology for this mistake and corrected the text in section 2.4.4. Estimation of Oral Buccal Ulcer healing.
Discussion
Lines 369 to 371: “….Moreover, we developed a biocompatible gel based on G. verum ethanol extract….” You did not develop, because you did not make any pharmaceutical study on the gel formulation. The authors confirmed this in lines 377-380, where they confirm the reason for carbomer use in their gel. You just used a predetermined and predefined Gel formula. So the wording here needs to be changed to address this fact.
Response: Thank you for this useful suggestion. In fact, we used as a gel base described in our previous study, however there is no available gel formulation in literature containing G. verum extract. We absolutely agree to remove the world formulation from the tittle since we did not perform pharmaceutical study on the gel formulation. Therefore, we suggested novel title, please see Title page.
Lines 372-374: I did not understand until now how the phyto-compound separation, identification, and quantification study was performed. This part needs more clarification in the methodology and results and needs more discussion here.
Response: Thank you to this comment. We made corrections in these parts according to reviewer’s comment. Please see Material and method, Results and Discussion section.
Conclusion:
This part needs more attention to summarize all the study findings and define its significant application.
Response: Thank you to this comment. We modified the conclusion along with the your suggestion.
Other general notes:
The English language needs attention very much, especially the expression and grammar.
Response: Thank you to this comment. The whole manuscript has been carefully checked and errors were corrected.
Reviewer 2 Report
In this manuscript, Vuletic et al. described the healing properties of GVL gel in a rat ulcer model. Such effect is backed up by visualization of the ulcer site and mechanistic study (redox & COX-2 levels). I think the article is well-written and I recommend publication after addressing some minor issues.
Line 97 Even if it's from a published protocol, the LC-MS method should be briefly described here.
Line 130 and 197 Remove the numbering in the protocol.
Line 205 Is secondary antibody used for IHC? What is the detection method?
Figure 5 should be mentioned in the text, even if there are no statistical significance.
Figure 4C Y-axis should be NO2-. Nitric oxide (NO) should not be confused with nitrite (NO2-)
Figure 5B - Data is missing (Day 10 GVL gel)
Figure 5C - Unit of GSH is wrong (should be nmol instead of U?)
Line 307 Numbering is wrong
All figures: What is the sample size n? It should be noted below each figure. I recommend using standard format of p value (* P ≤ 0.05, ** P ≤ 0.01, *** P ≤ 0.001, **** P ≤ 0.0001).
Author Response
Line 97 Even if it's from a published protocol, the LC-MS method should be briefly described here.
Response: Thank you to this comment. We expanded Material and methods section with detailed information referring to HPLC analysis.
Line 130 and 197 Remove the numbering in the protocol.
Response: Thank you to this comment. We removed the numbering according to your suggestion.
Line 205 Is secondary antibody used for IHC? What is the detection method?
Response: Thank you for this comment. We corrected this part in the methodology section. We described detailed immunohistochemical methodology for Cox2 staining and inserted references for our methodology (reference 19). Please see section methodology, part histologic analysis.
Figure 5 should be mentioned in the text, even if there are no statistical significance.
Response: Than you to this comment. We added Figure 5 in the text.
Figure 4C Y-axis should be NO2-. Nitric oxide (NO) should not be confused with nitrite (NO2-)
Response: Thank you to this comment. We apology for this typing mistake.
Figure 5B - Data is missing (Day 10 GVL gel)
Response: Thank you to this comment, we made a mistake while creating the Figure 5. We added the missing data. Please, see Figure 5.
Figure 5C - Unit of GSH is wrong (should be nmol instead of U?)
Response: Thank you for this comment. We corrected unit for GSH in Figure 5C.
Line 307 Numbering is wrong
Response: Thank you to this comment. We corrected the numbering.
All figures: What is the sample size n? It should be noted below each figure. I recommend using standard format of p value (* P ≤ 0.05, ** P ≤ 0.01, *** P ≤ 0.001, **** P ≤ 0.0001).
Response: Thank you for this useful comment. The sample size is five animals per group (n=5), we noted that below each figure and use suggested format of p value (* P ≤ 0.05 statistical significance between compared groups).
Round 2
Reviewer 1 Report
The authors addressed all my comments in a satisfactory way
I have no further comments